# Crack Initiation and Growth Behavior of HVOF Stellite-6 Coatings under Bending Loading

Behzad Sadeghi [1], Pasquale Cavaliere [2,*], Angelo Perrone [2] and Alessio Silvello [3]

1 Centre of Excellence for Advanced Materials Application, Slovak Academy of Sciences, Dubravska Cesta 9, 84511 Bratislava, Slovakia; behzad.sadeghi@savba.sk
2 Department of Innovation Engineering, University of Salento, Via per Arnesano, 73100 Lecce, Italy; angelo.perrone@unisalento.it
3 Thermal Spray Center CPT, Universitat de Barcelona, 08028 Barcelona, Spain; asilvello@cptub.eu
* Correspondence: pasquale.cavaliere@unisalento.it

**Abstract:** Stellite-6 powders were sprayed on Ni-Al bronze in order to produce coatings via high-velocity oxygen fuel (HVOF). The microstructural observations revealed the main mechanisms taking place for the substrate–coating adhesion. It was revealed that tungsten-rich particles are very active in improving the coating adhesion as well as the mechanical properties. The X-ray diffraction analysis of the coating material showed pronounced peak broadening, revealing high residual stresses related to excellent bonding to the substrate. As expected, the coating procedure led to an increase in surface hardness. The surface properties of the coatings were evaluated through cyclic three-point bending tests at different maximum loads. It was demonstrated that the main part of the fatigue life is spent in the crack initiation stage, with a short propagation stage. Obviously, this behavior decreases as the maximum cyclic stress increases. The micro-mechanisms taking place during cyclic loading were evaluated through fracture surface observations via scanning electron microscopy.

**Keywords:** Ni-Al bronze; Stellite-6; HVOF; 3-point bending; crack behavior

## 1. Introduction

Ni-Al bronzes are one of the most employed alloys in ship constructions due to their forming properties coupled with excellent mechanical and corrosion resistances [1,2]. In the past, many alloying elements (W, Ni, Al and Fe) were added to the base composition in order to improve these basic properties as well as to increase the fatigue resistance and fracture toughness in large ranges of temperature [3]. Given the high cost of these alloys, a repairing procedure through additive manufacturing via cold spray can lead to strong reductions in costs of maintenance [4].

In addition, many strong applications in sea water, especially at low temperatures, require the protection of Ni-Al bronze components through high-resistance coatings [5].

It is demonstrated that Co-based alloys deposited through thermal spray techniques can allow for the increase in the surface performances of such components. Nevertheless, given that the damage under service mainly starts form the surface, the soundness of the coatings under in-service conditions must be precisely defined [6–8]. In the case of repairing, the surface fracture behavior is fundamental for the definition of the coatings' performances in many conditions and for different materials [9–11]. Among cobalt-based alloys, Stellite 6 is well known for of its ability to improve many surface properties of metallic components [12–14]. The main properties of this alloy are recognized as its excellent stability in a large range of temperatures and its retention of high mechanical properties at high temperatures [15]. In addition, this alloy shows excellent resistance to fatigue and creep, as well as to corrosion at high temperatures [16–18]. During the coating procedure, Stellite 6 shows the formation of many intermetallic compounds able to improve the surface resistance of coated metallic components; nevertheless, the fracture

behavior at very thin levels must be well understood to apply this material for coatings and repair [19–26]. In fact, it is demonstrated that coating voids are favored in the presence of carbides and oxides [27]. This is confirmed for many different substrates [28]. The HVOF of stellite coatings is demonstrated to induce relevant residual stresses in the coating [29], residual stresses are at levels optimal for the sound fatigue properties of these kinds of coatings. Bond strength governs the fracture behavior of such kind of coatings [30,31]. In this peculiar field, very little scientific evidence is presented in the literature on the fracture behavior of Stellite 6 coatings on bronzes, especially if produced via HVOF [32–36]. In addition, very little experimental evidence is presented on the monotonic and cyclic fracture behavior of such kind of thermal spray coatings. Here, Stellite 6 powders were sprayed through HVOF on bronze substrates in order to evaluate the coatings' fracture behavior. Additionally, the micromechanisms acting during cyclic deformation are highlighted.

## 2. Experimental Procedure

Ni-Al sheets (50 mm × 20 mm × 5 mm) were cut form cast ingots. The surfaces were polished through sand blasting before coating operation. Nickel–aluminum–bronze disks were obtained by cutting cast ingots; the samples measured 30 mm in diameter and 5 mm in thickness, and they were used as the substrate. The sand blasting of the bonze specimens' surfaces was performed before HVOF operations. Stellite 6 powders with a nominal mean diameter of 53 μm were employed for the present study. The chemical composition in wt.% was: Cr-28.6; W-5.9; Ni-2.2; Fe-1.9; C-1.3; Mn-0.3; Mo < 0.1; Co-bal.

HVOF coatings were produced using Diamond Jet Hybrid DJH 2600 equipment (Oerlikon Sulzer Metco, Zürich, Switzerland). The oxygen gas pressure was 22 bar, the gas carrier rate was 4 L/min, the spray distance was set at 25 mm, the fuel feed rate was 280 L/min and the oxygen feed rate was 830 L/min.

XRD measurements were carried out by employing a X'PERT MPD diffractometer (Philips, Almelo, The Netherlands) with the following settings: constant power of 30 kV; constant current of 30 mA; monochromatic CuK$\alpha$ with a wavelength of 1.541 A; scan rate of 1 sec; step size of 0.03 degrees. Point X-Ray measurements were also employed to measure the coatings' residual stresses from the peak intensity and broadening. Microhardness measurements were performed by employing the Koopa microhardness instrument model MH3 (KoopaCo, Sari, Iran). During all the hardness tests, the force was 50 gr with 10 s dwell time. The coatings' microstructures were investigated by employing an Olympus optical microscope and a Philips XL30 SEM (Philips, Almelo, The Netherlands). The coatings' fracture behavior was analyzed through 3-point bending tests. The tests were interrupted once the coatings fractured. Cyclic bending tests at different maximum applied loads were performed in order to monitor the crack initiation and growth in the coatings under fatigue loading conditions. The tests were performed at a frequency of 1 Hz by employing an Instron 8801 standard mechanical testing machine (Instron, Norwood, MA, USA). The applied maximum bending loads were 3 and 4 kN, and the load ratio was set at R = 0.1 for all the tests. The crack growth rate was monitored by employing the direct current potential drop method [37].

## 3. Results and Discussion

### 3.1. X-ray Diffraction

The stellite powders employed in the present study are shown in Figure 1.

In Figure 2, the diffraction pattern of the employed particles and of the coatings produced via HVOF is shown.

A remarkable broadening of the peaks was observed in the case of the pattern on the coating, as expected for thermal sprayed materials. This is mainly due to the residual stresses induced in the coating by the particle deformation and cooling. Additional broadening is due to the grain refinement as a consequence of the spray process. The main phase is the face cubic-centered one of the cobalt.

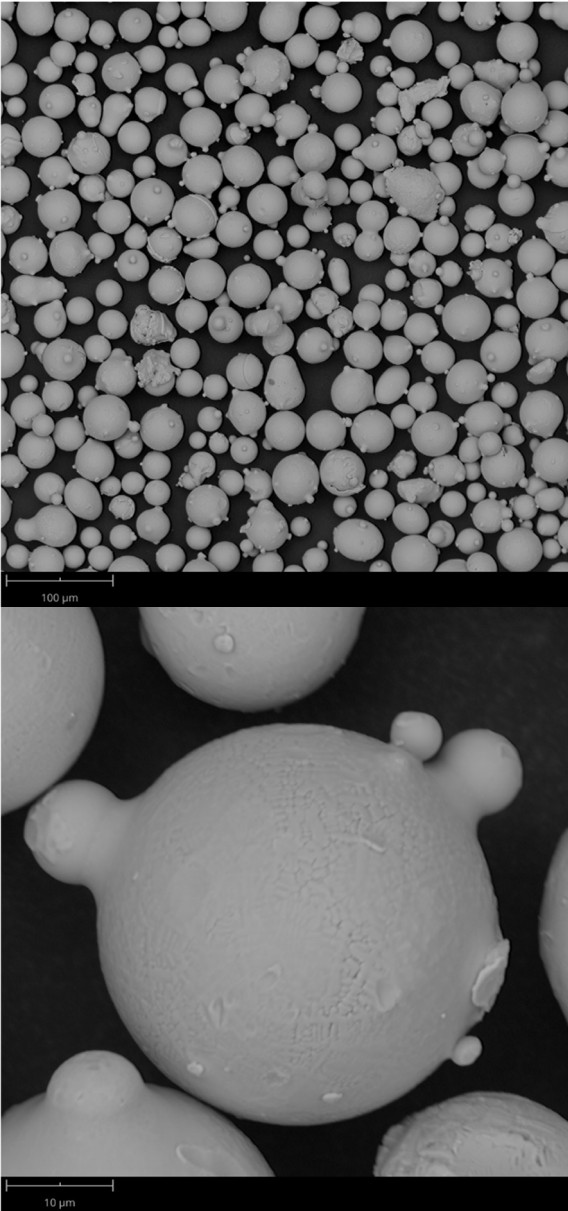

**Figure 1.** Stellite powders at different magnifications. The medium particle size was 30 microns.

The presence of carbides can be highlighted by the observation of the different peaks. Carbides were not revealed in the coating due to the severe plastic deformation and the cooling behavior of the material. In fact, carbide formation requires very slow cooling; this condition is far from that acting during the coating formation in HVOF. In fact, the fast cooling during HVOF acts as a retardant of the formation of carbides. The fast deformation and cooling can also act to induce nanostructuring in the deposited material [37]. Thermal spray processes such as HVOF and cold spray are characterized by the very high kinetic energy of the particles upon impact. This high-impact energy leads to the very fast dynamic recrystallization of the sprayed material, leading to strong grain refinement with respect to the microstructure of the feedstock particles [38].

Therefore, the particles deform with very high energy, leading to high residual stresses and, as a consequence, to the peaks broadening in the diffraction pattern. This is due to the change in the crystal lattice, which can be measured through Bragg's law, leading to the measurement and definition of the residual stresses along the coating thickness (Figure 3).

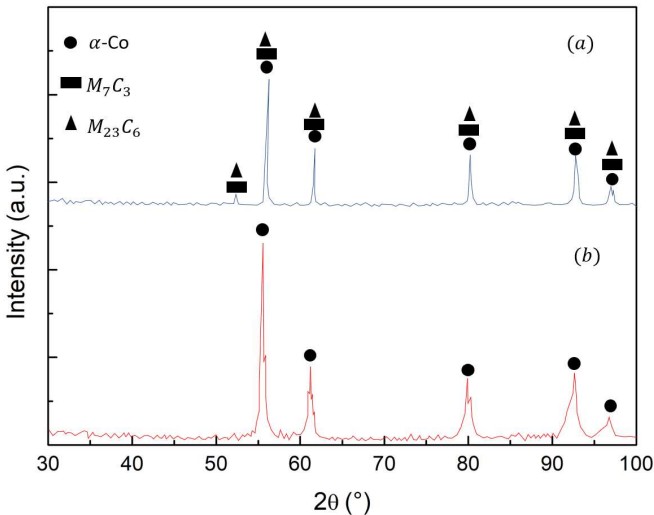

**Figure 2.** X-ray diffraction patterns of the employed powders (**a**) and of the sprayed coating (**b**).

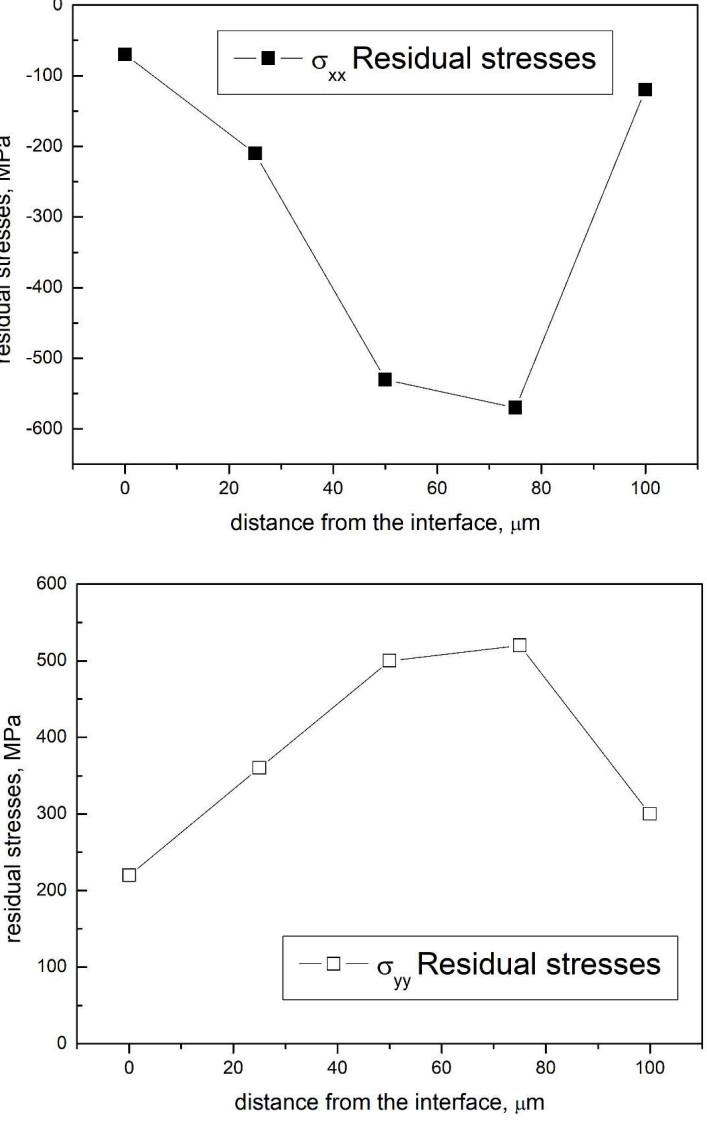

**Figure 3.** Residual stresses measured in the Stellite-6 coatings.

### 3.2. Microhardness

Figure 4 depicts the hardness profile along the coating cross section and the substrate.

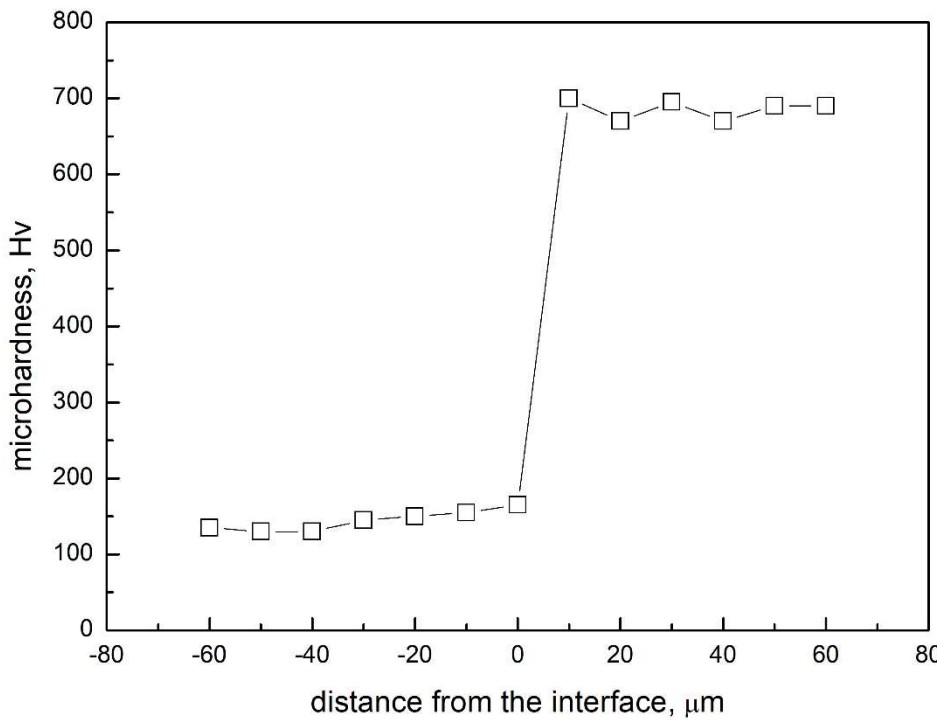

**Figure 4.** Microhardness profile of the cross section of the analyzed coatings on bronze substrate.

The substrate hardness was 120 HV. The coating procedure led to an increase in the surface hardness of up to 700 HV. This is due to the composition of the stellite particles and, in particular, to the presence of tungsten.

The hardness profile close to the interface is due to the work hardening of the particles impacting on the surface, also leading to increased residual stresses [32]. This is then amplified as the powders impact during the HVOF coating process [25,39,40].

### 3.3. Microstructure

The aspect of the cross section of the coating is shown in Figure 5.

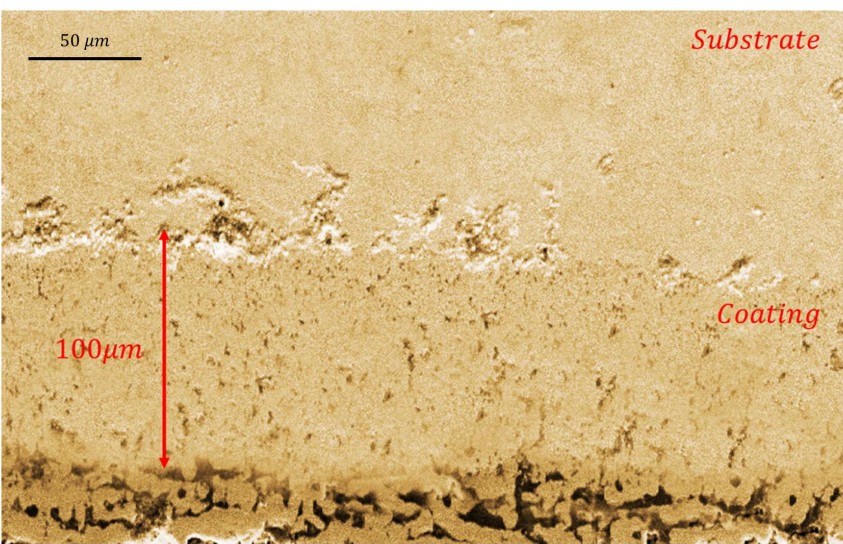

**Figure 5.** Coating aspect very close to the bronze surface.

The coating thickness appears to be very homogeneous, with a mean value around 110 µm. SEM observations revealed the splatted deformation of the particles and the different aspect of the interface due to the continuous deformation of the continuously impacting particles (Figure 6).

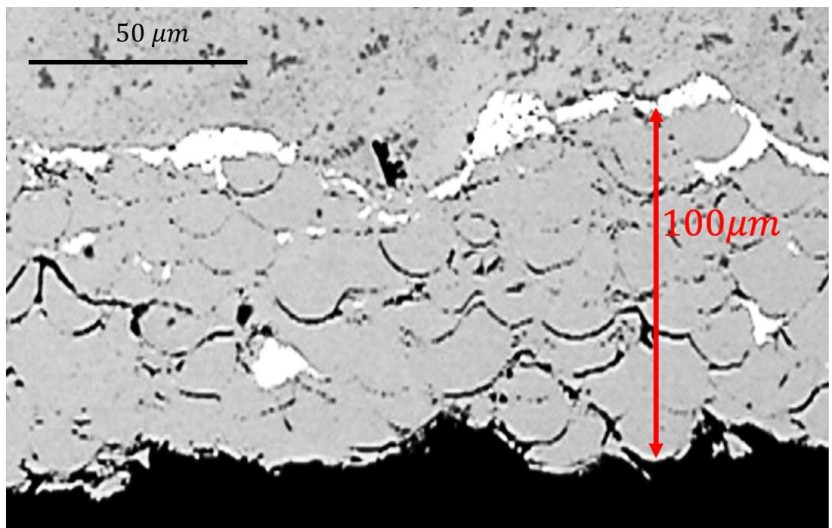

**Figure 6.** SEM aspect of the coating revealing the splatted particles.

XRD measurements of the white phases of Figure 6 revealed that the composition is (wt.%): W-87.42, Co-7.49, Fe-0.29 and Cr-4.79.

The coating porosity measured through Image J software was close to 1%. This very low porosity leads to the sound mechanical and corrosion properties of this kind of coatings. The white phases close to the interface were analysed thorugh EDS in the SEM; the compositional analysis is shown in Figure 7.

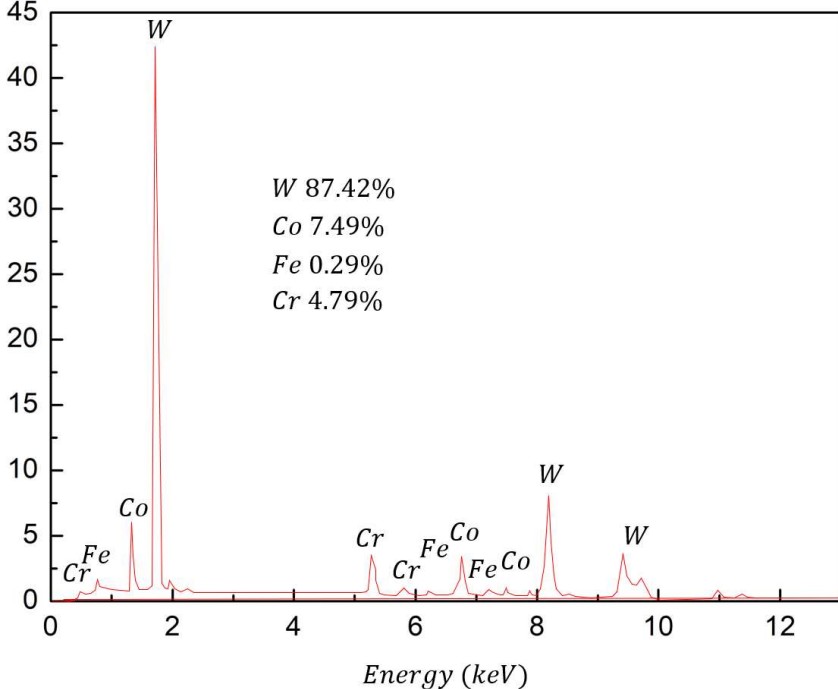

**Figure 7.** Point analaysis of the white areas of the coating surface.

The white zone are very rich in tungsten, this leads to a strong interface strength that is beneficial for the adhesion of the coating.

### 3.4. Bending Behavior

Theoretical and technological analyses of the fracture behavior of thermal spray coatings are fundamental to obtain fundamental information for the design and deposition of thermal spray coatings [41]. Figure 8 depicts the load–displacement curve for the coating specimen and substrate materials under the three-point bending test.

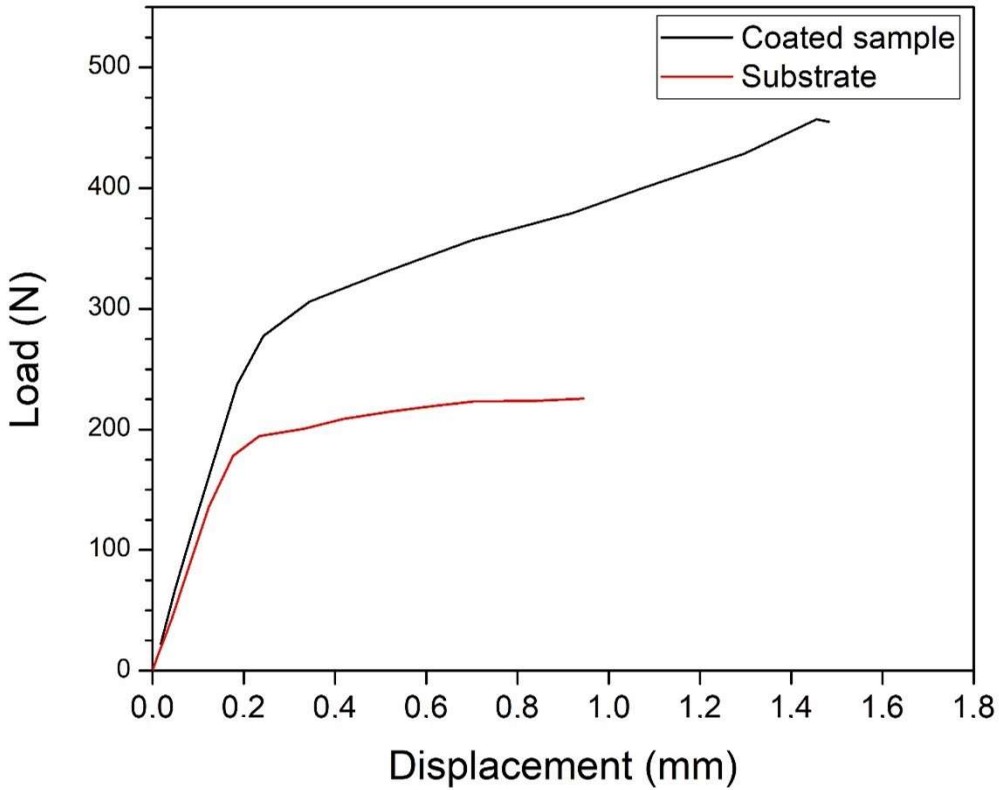

**Figure 8.** Three-point bending test of the HVOF coated samples.

Two regions of the curve of the Stellite-6 coating sample are highlighted in Figure 8. The first stage of the load–displacement curves revealed the elastic behavior of the substrate–coating structures. In this region, the stress had a linear relation with displacement when the displacement was less than 0.22 mm; generally, this region is called the elastic loading region. After that, once a critical force was reached (240 N for the coating and 170 N for the substrate), the elastic–plastic loading region developed. In this region, the stress increased with displacement, but its rate was much lower than that in the first stage. Once the displacement continued to reach 1.47 mm, cracks appeared, and subsequently, by continuing to apply the force, the cracks propagated rapidly towards the interface between the coating and substrate (Figure 9).

It was seen that the substrate materials had lower yield stress and a low work hardening capacity. The failure load for the coating sample was 453 N. The qualification of the deformation mechanisms belonging to the damaged structure, the applied force and the corresponding fracture behavior inside the coatings were crucial. The fracture surfaces of the coatings revealed that the rupture was ductile (Figure 10).

The surfaces are, in fact, characterized by a medium size and the presence of very fine dimples.

The fatigue crack properties of the deposited coatings are fundamental for the overall resistance of the materials where they are sprayed. This is a key issue for all those coatings designed and produced for strength increase, as well as for wear and corrosion protection. In fact, the tolerance to monothonic and dynamic loadings is related to the surface resistance to crack initiation and growth so determining the coatings' in-service life. In fact, it is crucial to gain precise information on the surface resistance to complex loadings in order to define

the fracture toughness and the deformation mechanisms in different stress and strain conditions. The subject is very complex; in fact, even if a high fracture resistance is required for the coatings' integrity, high strength is often accompanied with low fracture toughness. This aspect is crucial in thermal spray coatings; in fact, they normally show very high strength and low ductility due to particle decohesion as a result of the residual porosity. High strength and low ductility are factors that cause the fast degradation of the coatings, especially under cyclic loading [36]. Most of the fatigue life of a component was spent in the crack initiation stage, and the period of fatigue crack propagation was small. It is now widely accepted that fatigue crack initiation occurs early in life, and then the cracks grow through microstructural barriers. This growth period can occupy a considerable portion of the fatigue life, and it is essentially the fatigue crack propagation period [42]. Figure 11 shows the crack length as a function of the bending cycles for tests performed at maximum loads of 3 and 4 kN.

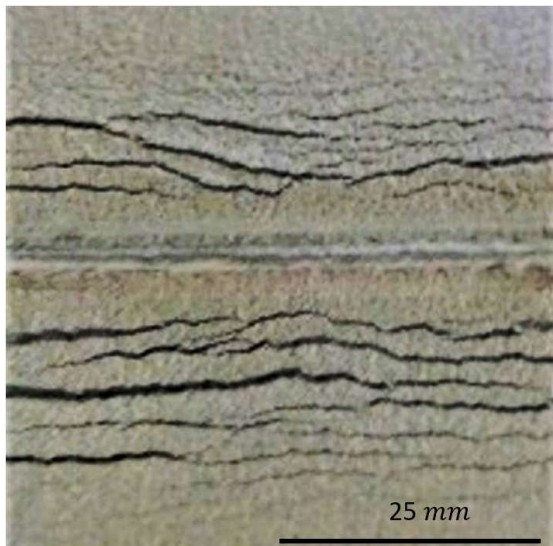

**Figure 9.** Cracks developed from the bending of the stellite coatings.

From Figure 11, it is clear how crack length slightly increased with the number of bending cycles in the first stages of loading. Close to 10.000 loading cycles, the relationship between the number of cycles and the crack length showed a different behavior with a sharper increase as the number of cycles increased. This is symptomatic of an increase in stress concentration in the front of the advancing cracks. As expected, the fracture behavior changed as the maximum bending load varied. All the fracture toughness evidence in coatings leads to very complex mechanisms observations because the fracture behavior depends on the properties of both the substrate and the coatings materials, as well as on their interaction upon loading. In this case, the coating–substrate interfacial issues, as well as the residual stresses, play a crucial role. The adhesion strength and the residual stresses, in fact, strongly influence the plastic zone development around the crack tip. These factors all influence the coatings fracture toughness related to the crack initiation and growth behavior. In the present case, the crack nucleated on the coatings' surfaces. Once nucleated, the crack grew and propagated toward the bulk. In general, in the case of high interfacial strength, the nucleated crack grew in the vertical direction. On the contrary, in the case of lower interfacial strength, the crack path changed when the coating delamination was produced. In this last case, the in-service life of the coating was very brief. Once local delamination occurred along the crack propagation, damage quickly developed, leading to a decrease in the rate of subsequent cracking appearance [43]. In the cases of flat specimens (without notches), several surface cracks appeared, with a few of these forming along the coating-substrate interface [44]. With the increase in bending loading, the normal strain at

the interface increased with a consequent formation of interface cracking. In some loading and propagation conditions, the cracks coalesce by producing the coating's spallation. The local delamination is also favored in the case of a fast increase in the crack density with consequent cracks coalescence. In the case of many superficial cracks, the density is directly proportional to the interface strength [45]. From the pioneering work of Paris et al. (1961), the rate of crack growth is dependent on a power relationship [46,47]:

$$\frac{da}{dN} = C(\Delta K)^m$$

where $a$ is the crack length; $N$ is the number of cycles; $C$ and m are the so-called Paris coefficient and exponent, respectively; and $\Delta K = K_{max} - K_{min}$ is the stress intensity range.

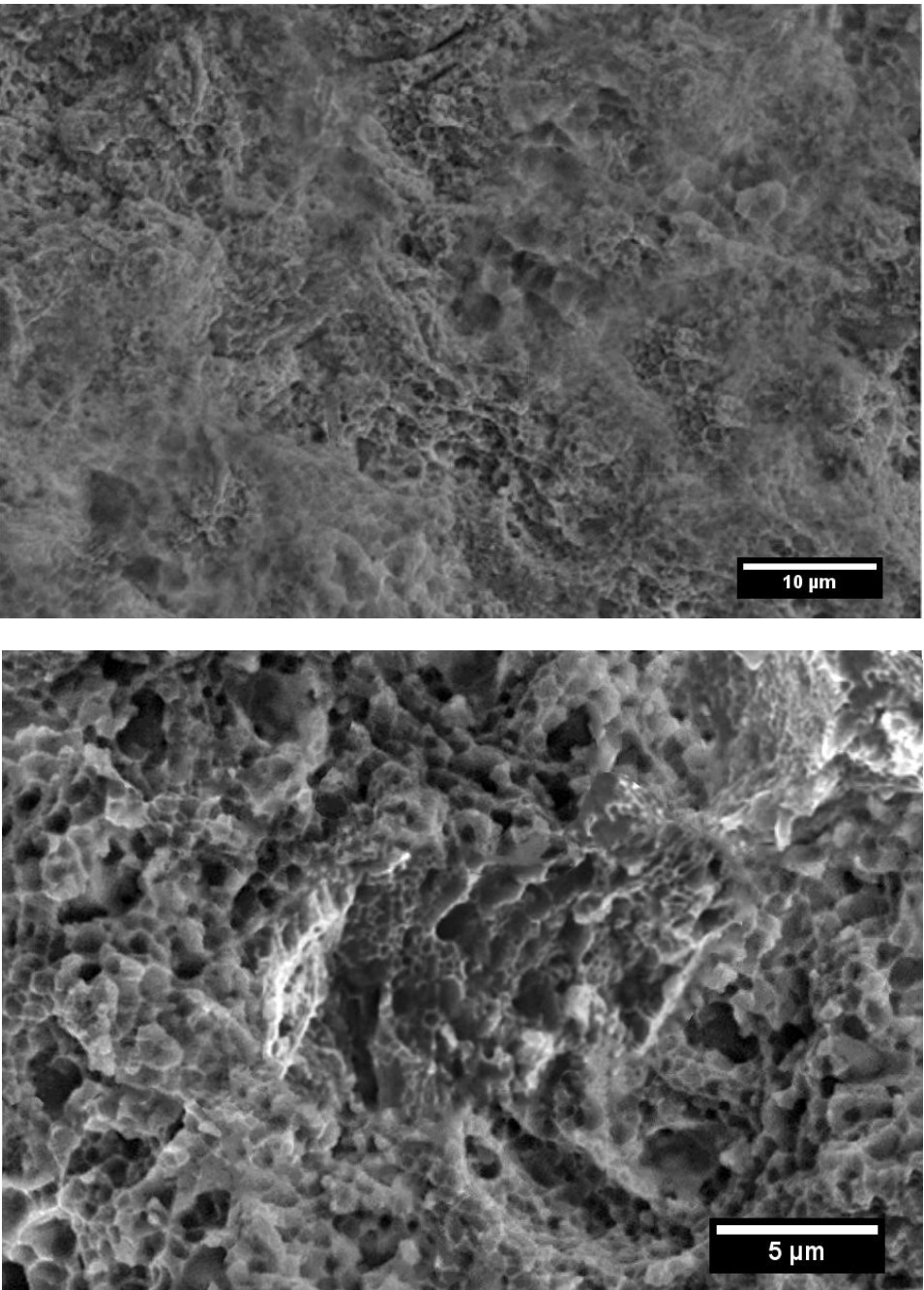

**Figure 10.** Fracture surfaces of the monotonic 3-point bending-tested coated samples.

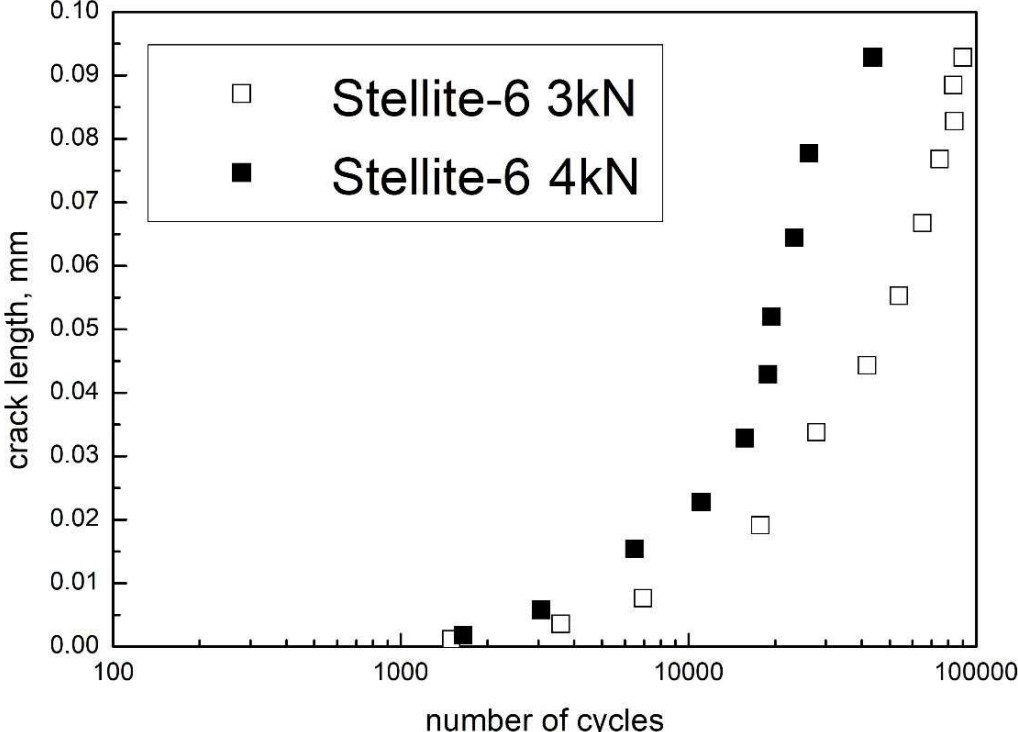

**Figure 11.** Crack length as a function of the number of cycles at 3 and 4kN maximum bending load.

Plasticity-induced crack closure is the contact of plastically deformed residual material in the wake of a fatigue crack. Upon loading, crack surfaces separate when the opening stress level is reached, and only then can the crack advance. Thus, fatigue crack closure modifies the stress intensity factor range. The Paris relation is widely accepted for describing the fatigue crack growth behavior of materials, and the correction for crack closure provides a more accurate description of the fatigue crack growth rate. By considering the effect of the crack closure during cyclic loading, the previous equation was modified:

$$\frac{da}{dN} = C\left(\Delta K_{eff}\right)^{m}$$

where $\Delta K_{eff} = K_{max} - K_{clos}$.

By measuring the crack length of the central main crack, the $\Delta K$ variation as a function of $da/dN$ was plotted (Figure 12).

Monotonic or cyclic loading of coated substrates led to the formation of cracks initiating and growing in the coating being thinner than the substrate. In the case of the tested samples, the crack stability was more pronounced for a maximum bending load of 3 kN with respect to the case of 4 kN of maximum bending loading. The crack propagated in the center of the sample without any apparent propagation at the coating-substrate interface for the 3 kN maximum load. The fracture propagated in the model in the coating material, no multiple cracks were detected and no additional macroscopic damages could be underlined. In addition, this propagation behavior allows us to highlight that the cyclic deformation mode of the coatings is completely different if compared to the monotonic bending. In the latter case, multiple cracking and coating–substrate interface decohesion occurred and, after a given strain, cracks tended to propagate along the interface.

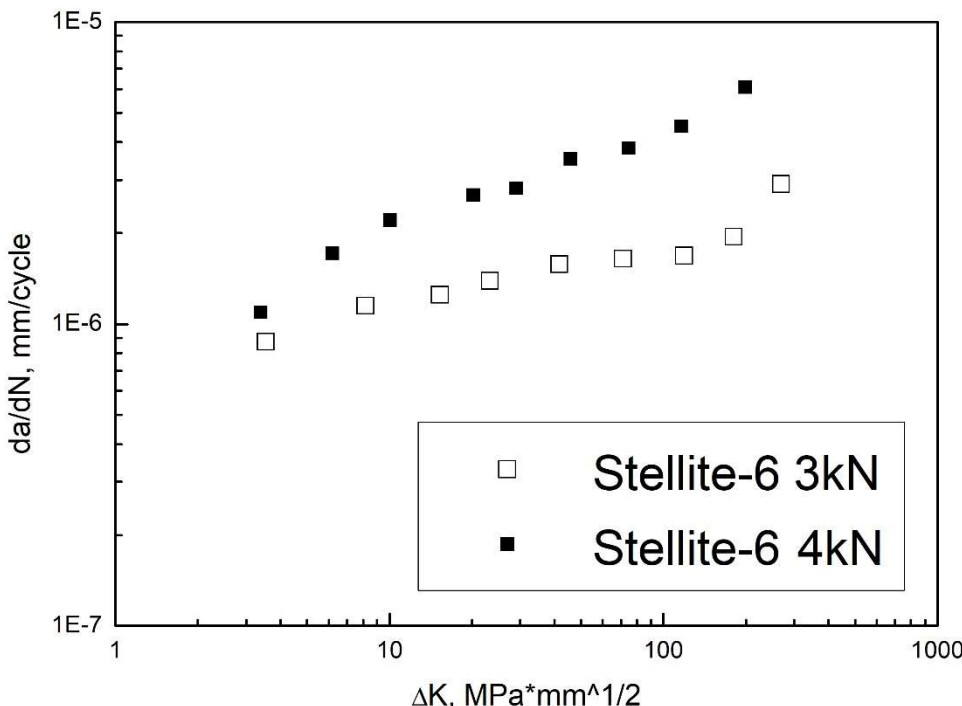

**Figure 12.** Crack growth rate as a function of the applied $\Delta K$ for all the tested samples. Format like 1E-6 means $1 \times 10^{-6}$, ^ means superscript.

## 4. Conclusions

- Cobalt-based stellite particles were employed to successfully produce coatings via HVOF on nickel–aluminum–bronze substrates.
- The coating procedure led to a very remarkable increase in the surface hardness.
- The monotonic bending of the coated samples revealed a ductile behavior with the appearance of many superficial distributed cracks on the surface of the coatings. The coating improved the strength.
- The ductility behavior was confirmed by the SEM of the fractured surfaces.

**Author Contributions:** All the authors contributed to the conceptualization, data, writing and revision. All authors have read and agreed to the published version of the manuscript.

**Funding:** This research received no external funding.

**Informed Consent Statement:** Not applicable.

**Data Availability Statement:** Not applicable.

**Conflicts of Interest:** The authors declare no conflict of interest.

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
