# Peer review of "Crack Initiation and Growth Behavior of HVOF Stellite-6 Coatings under Bending Loading"

_2674-0516, doi:10.3390/powders1020006_

Round 1

Reviewer 1 Report

The conclusions of each publication are very important! The conclusions must be supplemented by the basic presented results. The description of many presented measurement results is interesting, but insufficient. It is necessary to interpret the results of measurements in the context of the research aim. The aim of this work is to predict the crack behavior, which leads to an estimate of the residual life of the coating. The maximum number of cycles until fracture can then be estimated.

Of course, for estimation it is necessary to know the crack propagation rate under given bending loading. Crack growth rate – this "speed" – is determined by many factors, a quantity that allows a comprehensive description of their effect is the amplitude of stress intensity coefficient (see figure 11?). There are several models, but it must be explicitly stated in the text that this is, for example, the validity of the Paris-Erdogan law. Does this also apply for the coatings? This needs to be carefully explored in the future.

Author Response

The propagation laws were described in the text and also with additional references.

The applied low seems to work with those coatings not being thin films but medium thickness bulk coatings

Reviewer 2 Report

Dear authors,
The manuscript entitled 'Crack initiation and growth behavior of HVOF Stellite-6 coatings under bending loading' presents experimental studies on stellite-6 powders sprayed on Ni-Al bronze substrates in order to produce coatings via high velocity oxygen fuel. The main goal is to evaluate the coatings fracture behavior. Also, the authors looked onto the micromechanism acting during cyclic deformation are highlighted.
Although the idea is good, the manuscript needs major revisions. Thus, both the abstract and the introduction must be extended so as to include state-of-the-art in the field, respectively the main goal and the outstanding result of these studies.

Moreover, the techniques used to study these deposits (layers deposited on the respective support) must be explained in the methods section.

The manuscript can be considered for publication only after a major revision: MAJOR REVISION.

Author Response

-Abstract has been enriched by highlighting the main obtained results.

-Introduction has been enriched with the most recent references on stellite coatings.

-The needed coating procedure informations have been provided.

Reviewer 3 Report

In this work, crack initiation and growth behavior of HVOF Stellite-6 coatings under bending loading was investigated. However, some comments on the morphology of the coating were found to be at variance with the experimental results, and some statements made in the discussion section were found to be insufficiently supported by experimental data or analysis. The language needs to be carefully revised before publication, such as R98 “So. the particles deform With very high energy leading to high residual stresses and” , R194 “Once nucleated, crack grows and propagate toward the bulk.” and so on.

  1. To easily understand the experiment procedures, the detailed experimental method is necessary, especially the figures of the Stellite-6 powder and powder particle size distribution.
  • Figure 4, Figure 5 and Figure 8 do not have scales and all the figure of microstructure are of poor quality, it is recommended to provide them again.
  1. R112, the hardness of the surface increases. The author judges that it is the reason for the existence of tungsten. Please provide evidence to support it. What is the composition of the black area in Figure 6?
  2. R113 “ The hardness profile close to the interface is due to the pre-coating treatment of the substrate leading to strong superficial work hardening.” The author does not mention pre-coating treatment of the substrate in this article, and it is recommended to supplement.
  3. R206 “By measuring the crack length of the central main crack, the ∆K variation as a function of da/dN was plotted.” The author need to explain the ∆K, a, N and “mode I”of R205.

Author Response

-Figures of the powders were provided and their mean dimensions were indicated in the text.

-Figures 4, 5 and 8 were modified following the instructions.

-The composition of the phases was provided in the text.

-The sentence about the pre-coating treatment was modified.

-The crack propagation parameters were well defiend in the text.

Round 2

Reviewer 2 Report

The revised version of the manuscript is, in my opinion, suitable for consideration for publication in Powder, following the formatting rules imposed by the journal.

Reviewer 3 Report

For the presented modified version, I suggest that it can be accepted. 

This manuscript is a resubmission of an earlier submission. The following is a list of the peer review reports and author responses from that submission.